# Increasing temperature exacerbates acute toxicity effects of a glyphosate-based herbicide (Roundup) on the ostracod *Cypridopsis africana*

Yusuph A. Kafula*

## ABSTRACT

Predicted global temperature increase is expected to have unprecedented effects on freshwater ecosystems. While the individual impacts of temperature and chemical exposures on aquatic species physiology, life history and survival have received adequate attention, the interactive effects of temperature and chemical toxicity, particularly on the temporary pond ecosystems, have yet to be elucidated. Here, I conducted short-term toxicity tests to assess the impact of increased temperature on Roundup toxicity on an ostracod, *Cypridopsis africana*. Roundup is currently a widely used herbicide with glyphosate as its active ingredient. Test organisms were exposed to five nominal concentrations of Roundup (25, 50, 100, 200 and 400 mg/l active ingredient) and a control. These concentrations were crossed against two temperature conditions, 27°C and 31°C. After 48 h of exposure, Roundup lethal concentration 50 ($LC_{50}$) at 27°C was 223±15.2 mg/l and decreased to 186.03±4.04 mg/l at 31°C. These findings show a decrease in $LC_{50}$ by 16.5% following an increase of +4°C. This underscores the need for a tailored risk assessment paradigm that takes increasing temperature into consideration.

KEY WORDS: Glyphosate, Pollution, Safe concentration, Ecotoxicology, Ecological risk assessment

## INTRODUCTION

The dramatic decline in aquatic biodiversity is partly attributed to changes in temperature conditions, and, even at the predicted modest global warming of 1.5°C, substantial threats to and extinction of aquatic species have been observed (Porter et al., 2013). This biodiversity loss is, in turn, linked to a severely impacted socio-economic wellbeing of the people, among others because of dwindling fisheries resources, which are the affordable protein sources for most resource-poor communities (Ahamed et al., 2012). Temperature increase is, unfortunately, not the only threat to aquatic biodiversity. Agricultural intensification and use of pesticides pose additional threat that simultaneously impact aquatic ecosystems, especially in developing countries, where agriculture is rapidly intensifying (Garcia et al., 2018). Use of pesticides adjacent to aquatic ecosystems has adverse effects when chemicals drift by wind or surface runoffs to contaminate the ecosystems. For example, it has been established that only about 50% of sprayed pesticides reach targeted crops, with less than 0.5% eventually reaching the targeted pest (Pimentel and Burgess, 2012).

Roundup is the most commonly used glyphosate-based herbicide in Tanzania (Kafula et al., 2021). The herbicide is mainly used in agriculture and for aquatic weed control (Kafula et al., 2021; Ortiz-Ordoñez et al., 2011). Roundup is a trade name for a herbicide consisting of glyphosate as the main active ingredient and a nonionic polyethoxylene amine (POEA) surfactant to facilitate uptake in plants (Kafula et al., 2021; Hued et al., 2012). The mechanism of action in Roundup involves interfering with the production of essential aromatic amino acids (Kafula et al., 2021; Hued et al., 2012). Although the toxicity mechanism of Roundup in animals is not well understood, it is associated with high oxidative stress through the production of reactive oxygen species (Ortiz-Ordoñez et al., 2011).

It is now unequivocal that average global temperature will increase by over 1.5°C above pre-industrial levels in the next decade. According to the Intergovernmental Panel on Climate Change (IPCC, 2021), global temperature may further increase by +4°C by the year 2100. Changing air temperature has direct consequence on water temperature, which subsequently alters the oxygen solubility and pH, hence affecting the habitability of aquatic environments (Porter et al., 2013). These effects can be more pronounced in small water bodies such as temporary ponds. Moreover, metabolic processes and body functioning of all organisms depend on optimal temperatures, in which deviation from these temperatures necessitates adaptive mechanisms, which can be an added energy burden, while those failing to adapt go extinct (Garcia et al., 2018). Climate-related biodiversity loss in aquatic environments far outstrips that of terrestrial environments at the predicted modest global warming of 1.5°C (Kingsford et al., 2016; Garcia et al., 2018).

Temporary ponds (also called pans, playas or ephemeral pools) are small bodies of water, generally <10 ha large and are characterized by periodic droughts (Kafula et al., 2023; Waterkeyn et al., 2008). Temporary ponds occur throughout the world and are used as model ecosystems for larger permanent freshwater ecosystems (Blaustein and Schwartz, 2001). Although small and often overlooked, temporary ponds are home to unique biota adapted to live in these harsh and often unpredictable habitats (Brendonck and Williams, 2000; Harris et al., 2002). Permanent inhabitants of temporary ponds – such as large branchiopods, water fleas, rotifers and ostracods – survive dry periods by producing drought resistant life stages (e.g. dormant eggs) that can stay viable for years (Blaustein and Schwartz., 2001; Griffiths, 1997; Bird et al., 2019; Brendonck and De Meester,

Department of Aquaculture and Fisheries, College of Agriculture, Mwalimu Nyerere University of Agriculture and Technology, P.O. Box 976, Musoma, Tanzania.

*Author for correspondence (yusuph.kafula@mnuat.ac.tz)

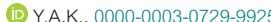 Y.A.K., 0000-0003-0729-9925

Biology Open

2003). As an adaptation to the often short-lived nature of their habitat, permanent inhabitants of temporary ponds display rapid hatching and fast growth and maturation rates, which enable them to complete their life cycle before ponds dry out again (Brendonck et al., 2022).

Although the impacts of pesticide contamination on aquatic organisms have received adequate attention from researchers, how temperature interacts with pesticides to drive the sensitivity of temporary pond species has yet to be elucidated. To further enrich the ecotoxicological dataset and contribute to aquatic species protection, I conducted toxicity tests to assess the role of temperature as a driver for Roundup toxicity on an ostracod, *Cypridopsis africana*. Due to an additional energy burden linked to species survival in temporary ponds, I hypothesized that the sensitivity of *C. africana* to Roundup will be exacerbated by increasing temperature.

## RESULTS

After 48 h of exposure to Roundup at 27°C and 31°C, lethal concentration 50 ($LC_{50}$) values were estimated by dose-response modelling and reported values do not correspond to experimentally tested concentrations. At 27°C, Roundup $LC_{50}$ was 223±15.2 mg/l active ingredient (a.i.) (Table 1, Fig. 1A and Fig. 2); however, following an increase in temperature by +4°C, *C. africana* sensitivity increased, leading to a decrease in $LC_{50}$ value to 186.03±4.04 mg/l a.i. at 31°C (Table 1, Fig. 1B and Fig. 2). Moreover, temperature increase and Roundup exposure acted individually to significantly reduce *C. africana* survival (Table 2). The interaction between the two stressors also had significant effect on *C. africana* survival (Table 2). A significant interaction between temperature and Roundup concentration indicates a steeper mortality response at 31°C (Table 2).

## DISCUSSION

Short-term toxicity of Roundup on *C. africana* was assessed under two temperature scenarios (24°C and 31°C). Test organisms were exposed to five different nominal concentrations of Roundup ranging from 25 to 400 mg/l. At 27°C Roundup had an $LC_{50}$ of 223±15.2 mg/l a.i. and decreased to 186.03±4.04 mg/l a.i. at 31°C, reflecting a difference of 36.97 mg/l (16.6%) following an increase of +4°C.

Compared to an established invertebrate model, *Daphnia magna*, sensitivity of *C. africana* to Roundup is low. *D. magna* has been previously reported to have a Roundup $LC_{50}$ of 25.5 mg/l; this further increased to 962 mg/l when only the active ingredient (glyphosate) was tested, confirming that the toxicity of Roundup is highly influenced by the surfactant, POEA (Tu et al., 2001). Tolerance of *C. africana* may additionally be linked to adaptive mechanisms, enabling survival in temporary ponds. Permanent inhabitants of temporary ponds are adapted to living in highly variable habitats, with core water quality parameters often exceeding threshold levels for other aquatic organisms. Temporary ponds are characterized by rapidly changing pH, dissolved oxygen and dissolved solids due to dynamic water temperature. High water temperatures in temporary ponds often result from the absence of a temperature buffer

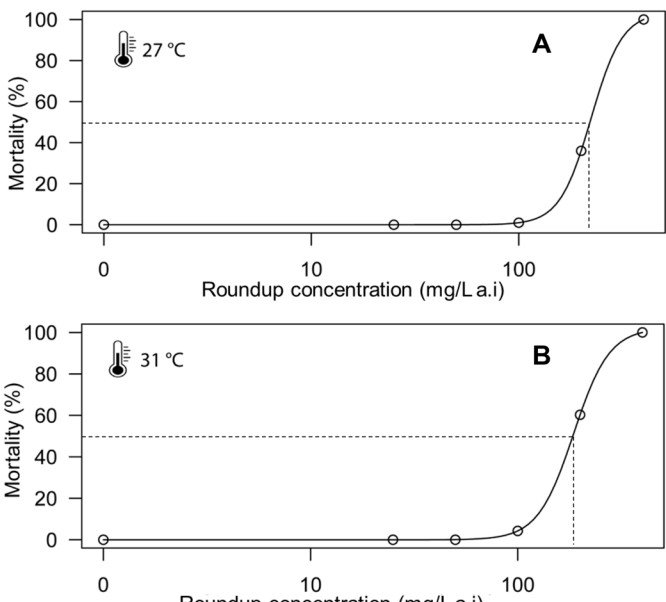

**Fig. 1. Dose-response curve showing cumulative mortality of *Cypridopsis africana* after 48 h of exposure to Roundup at 27°C and 31°C.** (A) 27°C. (B) 31°C.

due to their small area and shallow depth. However, comparative sensitivities of other temporary pond species e.g. fairy shrimps (*Streptocephalus* spp.) have been reported to be higher than established invertebrate models (Kafula et al., 2023). This further highlights individual variations among species and the need to generate more species-sensitive data to be able to tailor our chemical ecological risk assessments to adequately protect all species.

As I hypothesized, increasing temperature by +4°C exacerbated *C. africana* sensitivity to Roundup. We found significant interaction between temperature and Roundup in reducing survival of *C. africana*. Higher mortalities may be explained either by increased chemical absorption at higher temperatures or by higher metabolism, which leads to the production of toxic metabolites such as aminomethylphosphonic acid (AMPA). Alternatively, there could be a tradeoff of energy available to counteract the toxicity of Roundup by ensuring homeostatic state induced by an increasing temperature. This agrees with Kim et al. (2010), who found increased sensitivity of *D. magna* to pharmaceuticals, acetaminophen, enrofloxacin and chlortetracycline attributed to increasing water temperature. Increasing sensitivity was associated with altered toxicokinetics of chemicals and physiological mechanisms in *D. magna* (Niinemets et al., 2017; Kim et al., 2010). The role of temperature in driving chemical toxicity has been well documented, e.g. by Heugens et al. (2001), Pinheiro et al. (2021) and Li et al. (2014), underscoring the increase in chemical sensitivity of aquatic organisms following an increase in temperature. This further underscores the need to re-define the presumed safe concentrations, taking into account the envisaged global temperature increase.

Interaction between Roundup and temperature increase may exacerbate *C. africana* sensitivity through enhanced chemical absorption. Metabolic rates of aquatic invertebrates increase with an increase in temperature, potentially enhancing the absorption and bioavailability of Roundup (von Fumetti and Blaurock, 2018). This further implies that organisms in small and variable habitats such as temporary ponds may be more vulnerable to toxic effects under a changing climate. Similarly, higher temperatures can induce thermal

**Table 1. $LC_{50}$ values at 48 h after the start of exposure to Roundup under two temperature scenarios (27°C and 31°C), including the corresponding standard errors**

| Temperature condition | Duration (h) | $LC_{50}$ (g/l) | Standard error |
|---|---|---|---|
| 27°C | 48 | 223.01 | 15.19 |
| 31°C | 48 | 186.03 | 4.04 |

Biology Open

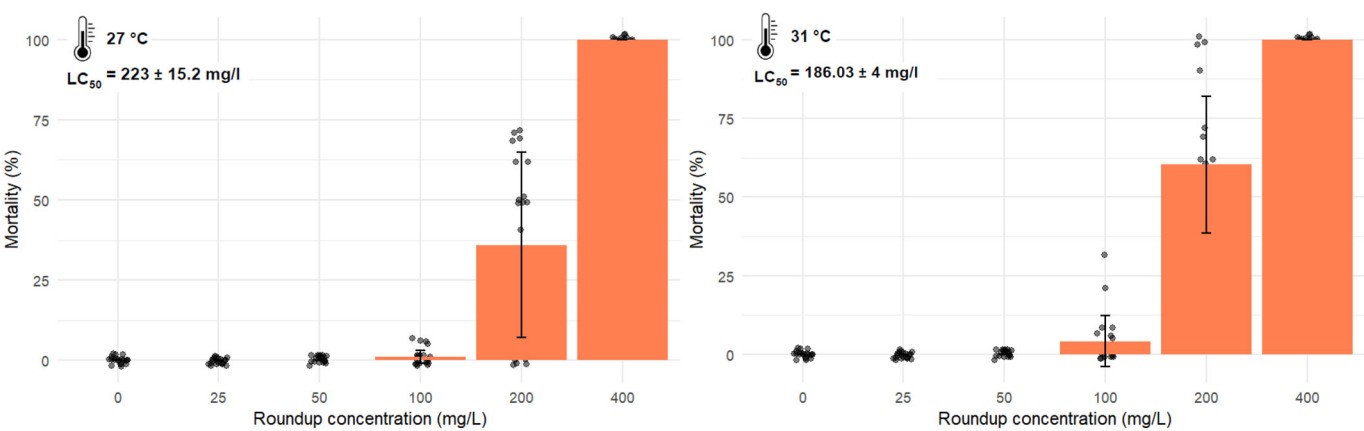

**Fig. 2. Ostracod mortalities across Roundup concentration gradient (0, 25, 50, 100, 200 and 400 mg/l a.i.), with corresponding LC$_{50}$ values at 27°C and 31°C.**

stress, leading to altered metabolic processes. Glyphosate, the active ingredient in Roundup, has been reported to affect energy production pathways, adding further stress and potentially leading to higher mortality (Strilbyska et al., 2022). In relation to the short-lived nature and fluctuating conditions (temperature, pH and dissolved oxygen) in temporary ponds, organisms invest in developing mechanisms to cope by increasing their growth rates and efficient respiration. This causes a tradeoff between survival mechanisms in temporary ponds and resistance to chemical exposure, rendering them more sensitive to Roundup.

## Conclusion
I showed that the toxicity of Roundup in *C. africana* is exacerbated by increasing temperature within the predicted global temperature increase in the next decade. The study therefore sets a scene and calls for further studies on possible synergistic effects of temperature and other chemical mixtures in our aquatic environments for tailored risk assessments and biodiversity conservation framework formulation. I additionally suggest further research into other commonly used pesticides using relevant endpoints to re-define the safe concentrations of these chemicals under increasing global temperature scenarios.

## MATERIALS AND METHODS
### Ostracods hatching and maintenance
Ostracods (*C. Africana*) were selected as the test species due to their key ecological role as filter feeders, their wide occurrence in temporary ponds, and their ease of breeding and maintenance in the laboratory. Ostracods were hatched from an integrated dry sediment sample with dormant eggs. Sediments were collected from five temporary ponds in the Lake Victoria Basin. Sampled ponds were located further away from agricultural areas to avoid possible previous exposure of test animals to Roundup. Deeper areas were identified prior to collection of sediments for dormant eggs. Then,

sediments were sampled from four established transects that radiate from the center (Kafula et al., 2023). Sediments were collected from the upper 3 cm as most viable dormant eggs are found in the upper 2 cm of the pond bottom (Kafula et al., 2023; Boven et al., 2008). Sediments with dormant eggs were air dried, homogenized, wrapped in aluminum foil and stored in the dark at room temperature before being evaluated. Sediment samples were screened by means of sugar floatation to select samples with high abundance of *C. africana* eggs (Pinceel et al., 2017). Then, hatching and maintenance of ostracods was done (with slight modifications) as described by Thoré et al. (2021). Briefly, 3 kg of mixed sediment sample were inundated in each of the five plastic containers with 140 l dechlorinated tap water at 27°C under a 14:10 h light:dark regime. Individuals were fed *ad libitum* with *Acutodesmus obliquus* (CCAP 276/3A). In two of the five hatching containers, temperature was gradually increased (using thermostat heaters) to 31°C for 48 h to allow for the acclimation of the test species prior to full scale toxicity tests [Organisation for Economic Cooperation and Development (OECD), 2004].

### Preparation of exposure media
Roundup was purchased, in liquid form, from a local pesticide shop in Musoma, Mara, Tanzania available as Roundup 360 SL (Monsanto, Bayer Agriculture BVBA, Belgium), which has pure N-phosphonomethylglycine – glyphosate (74.70%) as active ingredient and POEA (25.30%) as surfactant. A stock solution of 1 g/l Roundup was prepared and stored at −20°C. Experimental medium was produced by adding pesticide stock solution to reconstituted water to make five nominal concentrations of Roundup 25, 50, 100, 200 and 400 mg/l a.i. Reconstituted water was prepared by adding standardized salt (Instant Ocean Sea Salt, Instant Ocean-Aquarium Systems, Fiji) to distilled water to a conductivity of 490 μS/cm, matching the conductivity levels of temporary ponds.

### Acute exposure test experimental setup
Before conducting full-scale toxicity tests, range-finding tests were carried out to ascertain nominal concentration ranges that should be included in the experiment. The acute toxicity test ran for 48 h as described in the standardized protocol (OECD, 2004). A total of 1200 *C. africana* juveniles (96 h old) were divided into five experimental groups at different concentrations of Roundup and a control (where no Roundup was added) in each of the two temperature conditions (27°C and 31°C) (see Fig. 3). Temperature condition studied include the ambient water temperature in temporary ponds (27°C) and a projected increase of 4°C by the year 2100 (New et al., 2011). A group of five *C. africana* juveniles was transferred to a 25 ml jar, and each test concentration was replicated 20 times (Fig. 3), making a total of 100 individuals per treatment. Prior to the experiment, individuals were fed live *Acutodesmus obliquus* (CCAP 276/3A). However, to avoid introducing confounding factors, test organisms were not fed during the 48-h exposure duration (OECD, 2004). Basic water-chemistry parameters were measured throughout the experiment. Mortality was scored once daily and was defined as an individual being motionless for 15 s after

**Table 2. Output of generalized linear mixed effect models using mortality as a response variable and temperature, Roundup and their interactions as predictors**

| Predictors | Response (mortality) | | |
| --- | --- | --- | --- |
| | Estimate | z-value | P-value |
| Roundup | 0.056 | 52.16 | 0.0001 |
| Temperature | 0.28 | 255.98 | 0.0001 |
| Roundup*temperature | −0.57 | 0. 89 | 0.0001 |

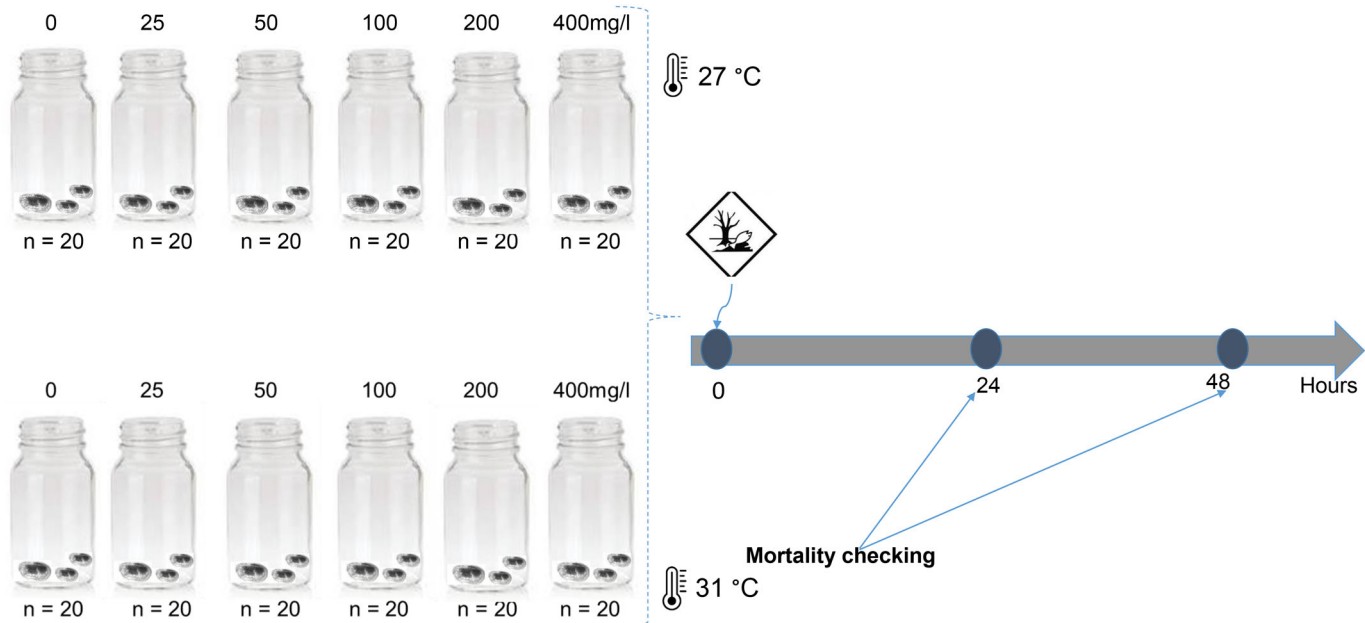

**Fig. 3. Acute toxicity experimental design showing two temperature conditions (27°C and 31°C) and five Roundup nominal concentrations used in the study.**

gentle agitation of the test vessel, based on OECD (2004). Nominal concentrations used were 0 (control), 25, 50, 100, 200 and 400 mg/l ai. Realized concentrations were 25.2±0.78, 49.87±0.05, 101.3±0.25, 199.78±0.85 and 398.92±0.55 mg/l of Roundup. Throughout the experimental duration, dissolved oxygen saturation levels were >80% in the experimental medium. Electric conductivity was 460±16.8 µS/cm, water pH was 6.4±0.8, and water temperature was maintained at 27±1°C or 31±1°C using thermostats in water tubs. During an experiment, survival was 100% in the control conditions both at 27°C and 31°C (Fig. 2).

### Realized concentration determination
Realized experimental media concentrations were measured using a High Performance Liquid Chromatography (HPLC) system (Model Wufeng LC 100, China) using a protocol modified from Qian et al. (2009). Briefly, the HPLC machine had two LC-10ATvp pumps and an SPD-10Avp. For separation, a reversed-phase Kromasil ODS C18 column (250 mm×4.6 mm inner diameter, particle size 5 µm) was used. Prior to HPLC analysis, samples were centrifuged to remove solid particles, and the supernatant was filtered through a 0.45 µm membrane filter. An aliquot of 1 ml of composite glyphosate standards was transferred to a vial, and pH was adjusted to 9.5. 2 ml of 4-chloro-3,5-dinitrobenzotrifluoride was added, and the mixture was mixed vigorously. Then, the mixture was heated in a water bath for 30 min at 60°C. We then added 50 µl of 2 M HCl and subsequently filtered the mixture through 0.45 µm nylon filters and injected it into the chromatographic system.

### Quality assurance and quality control measures
At each pesticide nominal concentration, at least three samples were analyzed. We also ensured that glassware used for sample processing and analysis were thoroughly washed and dried an oven at 150°C then allowed to cool in dust free cabinets. In the analysis, blank samples were used, and each sample was analyzed in triplicate to ensure quality. In all blank samples included in the analysis, no glyphosate was detected. Calibration curves were run to ensure that the correlation coefficient was >0.98. The minimum limit of detection was 0.01 mg/l.

### Data analysis
Data analysis was done in R v4.4.2. Mortality was scored as either 0 (dead) or 1 (alive). Generalized linear mixed effect models with binomial error distribution suitable for binary outcome data was used to predict the significance of Roundup exposure, temperature and their interactions (fixed factors) on *C. africana* survival. Jar ID was added in the models as a random factor. For generalized linear mixed effect models we used lme4 (Bates et al., 2015), car (Fox et al., 2012) and MASS (Ripley et al., 2013) packages. In the acute toxicity tests, $LC_{50}$ values were calculated from pesticide dose-response curves at 48 h in which mortality was used as response variable (Ritz et al., 2015), using the 'drm' function in the drc package.

### Acknowledgements
I thank Mr Erickson J. Kihundwa for his assistance in the laboratory and during wild collection of test species and sediments. I also thank Mwalimu Nyerere University of Agriculture and Technology for its kind contributions, including access to office and laboratory space, equipment and consumables, and vehicles for sampling campaigns.

### Competing interests
The author declares no competing or financial interests.

### Funding
This work did not receive any financial support. Deposited in PMC for immediate release.

### Data and resource availability
All relevant data and details of resources can be found within the article and its supplementary information. Datasets, methods and analytical approaches will be made available to all researchers upon request.

### Peer review history
The peer review history is available online at https://journals.biologists.com/bio/lookup/doi/10.1242/bio.062487.reviewer-comments.pdf

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
