## [Peer Review File · Biology Open]

Increasing temperature exacerbates acute toxicity effects of a glyphosate-based herbicide (Roundup) on the ostracod *Cypridopsis africana*

Yusuph Aron Kafula

DOI: 10.1242/bio.062487

Editor: Kendra Greenlee

Review timeline

Original submission:	13 January 2026
Editorial decision:	19 January 2026
First revision received:	10 February 2026
Editorial decision:	19 February 2026
Second revision received:	24 February 2026
Editorial decision:	27 February 2026
Third revision received:	3 March 2026
Accepted:	3 March 2026

Original submission

First decision letter

MS ID#: bio.062487

MS Title: Increasing temperature exacerbates acute toxicity effects of a glyphosate-based herbicide (Roundup) on the ostracod *Cypridopsis africana*

Authors: Yusuph Aron Kafula

I have now reached a decision on the above manuscript.

The reviewer reports are shown at the bottom of this email.

As you will see, the reviewers raised a number of substantial criticisms that prevent me from accepting the paper at this stage.

They suggest, however, that a revised version might prove acceptable, if you can address their concerns. In particular the concerns about proper controls, statistical analysis and reproducibility need to be adequately addressed. If you think that you can deal satisfactorily with the criticisms on revision, I would be pleased to see a revised manuscript.

At this stage, we also ask you to ensure your manuscript complies with our formatting guidelines. Provided you are able to fully address the referees' comments, we are positive about publication of your paper (we accept over 95% of revision submissions) and therefore hope you won't mind any extra work involved in reformatting your manuscript at this point.

Please upload both a 'clean' version of your Word file, along with a highlighted version clearly showing where you have made changes in the revised manuscript. Please avoid using 'Track changes' in Word files as these are lost in PDF conversion.

I should be grateful if you would also provide a point-by-point response detailing how you have dealt with the points raised by the reviewers in the 'Response to Reviewers' box. Please attend to all of the reviewers' comments. If you do not agree with any of their criticisms or suggestions please explain clearly why this is so.

Reviewer 1

Comments for the author

In this study, Dr Kafula conducted a laboratory investigation into how temperature (+ 4 °C) impacts toxicity to a compound called Roundup (trade name) in an ostracod from temporary pond systems in East Africa. Roundup toxicity was assessed at 27 °C and 31 °C by measuring LC50 over 48 hours. Results reveal lower LC50 values at higher temperature suggesting environmental warming associated with climate change increases the susceptibility of this species to Roundup toxicity. The author suggests that more research is needed on a range of species and chemicals to better understand chemical toxicity in a warming environment.

I believe this research is timely and important and I think this case is well made by the researcher. I also find the conclusions to be somewhat well justified by the data, with caveats (see below). The written English is generally good, but I found a limited number of serious grammatical errors which impair reader understanding and need rectification. There are, I feel, several fundamental flaws with how the study was conducted and analysed statistically, and some key information is missing from the methods and discussion. The main issues I have with this study are:

1) The hypothesis is not fully justified: At the end of the introduction the author states that they expect increasing temperature to increase Roundup toxicity, but the rationale for this expectation is missing. Why is the hypothesis that it will increase toxicity and not decrease? What has other research shown, and why does the author hypothesise this direction?

2) Terminology of Roundup vs glyphosate - these terms are used interchangeably throughout the title, abstract, introduction, methods and figures and should be kept consistent. Given the statement in the discussion about Roundup vs glyphosate toxicity (L215), I would strongly suggest sticking to the term 'Roundup' throughout.

3) Methods missing description of temperature control - There is no explanation for how temperature was controlled, manipulated or measured in the study, and crucially, no temperature data from the experiment are reported. Additionally, no explanation is given as to why + 4 °C was used as the high temperature treatment. Might this already be a lethal temperature for this species? Or is it within the temperature range found in temporary ponds?

4) Experimental design and statistics - This in my opinion is the biggest flaw in the study. The author does not employ a fully crossed experimental design (independent warming treatment, independent chemical treatment and the two combined equalling 4 treatments). As such there is no true control for temperature and it is not known whether the increased mortality at high temperature is due solely to temperature, or an interaction between temperature and chemical, potentially calling the findings and conclusions into question. As a result, the chosen statistics are invalid, and the individual effects of temperature and chemical cannot be statistically determined. This makes statistical results in table 2 void. The author should either repeat the study with a true temperature control or change the statistical analyses to a 1-level mixed effects model with warming/chemical exposure as the treatment.

5) Mechanistic explanation missing in the discussion - The discussion would greatly benefit from a deeper explanation for possible biological or physiological processes behind the findings as well as the toxicity mechanism of the chemicals present in Roundup. This information is key for the discussion to stimulate further thought and debate among readers, particularly those such as myself who are not familiar with the toxicity mechanism of this specific chemical.

Given these issues, I do not recommend that the manuscript be accepted in its current form. More detailed comments are given below.

Abstract

L33 and 39 - It is not clear to me what is meant by 'temporary' pond ecosystem in this sentence. This should be briefly defined in the abstract, or this detail omitted in the abstract.

L35 - Roundup should be briefly defined and explained in the abstract.

Introduction

Well written and clearly lays out the research question and its importance and timeliness. What is missing however, is why the author hypothesises temperature to increase sensitivity to chemical exposure. Has this been done before? Or is this the first time that chemical exposure LC50 has been determined in an invertebrate at different temperatures? (I highly doubt that).

L55 - Grammar: remove the word 'in'.

L72 - add the word 'with' after the word 'interfering'.

L78 - Stating that rising temperatures alter the water quality is not always strictly true. I would suggest re-phrasing this sentence.

L100 - Grammar: change 'elude' to 'eludes'.

L105-106 - The manuscript would benefit from more robust justification for the hypothesis. Why would LC50 be higher at high temperature? Has this been investigated in other species, other chemicals or other systems?

Materials and Methods

In general, the methods are clear and comprehensive, and the level of replication is strong, but some important details and justifications are missing as well as water composition data I believe to be relevant. Unfortunately, I am not convinced by the experimental design and accompanying data analysis. The statistical analyses (and results) presented suggest this experiment is a fully crossed design (eg temperature + chemical + chemical * temperature), which it is not. My understanding is that the author has only 2 treatments: a 48-hour LC50 test at 27 degrees (Fig. 2), and a 48-hour LC50 test at 31 degrees (Fig. 3). This means it is not possible to statistically deduce the isolated impacts of temperature on LC50 independent from chemical exposure. I would expect to see a treatment where binary mortality was recorded at both 27 and 31 degrees with no chemical exposure which would constitute a true control and allow the individual effects of temperature and chemical exposure to be statistically analysed as well as their interaction. I concede that conducting an LC50 test without a chemical present is tricky, but this is important to know if the 4 °C increase in temperature is causing the higher mortality by itself, or whether there is indeed a true interaction between temperature and chemical.

There is also no description of how temperature was manipulated and controlled in this study. Were experiments conducted in stable, temperature-controlled rooms? Or in water baths heated using immersion heaters? Also why was + 4 °C degrees of warming chosen? Is this based on IPCC end of century predictions or based on previous work?

L110 - Was the chemical purchased as a solid or solution?

L114 - Why were these concentrations chosen and how do they match with observed concentrations in the field or previous research?

L116 - I would suggest also including the salinity of this water (e.g. in ppt) for reference. Furthermore, the ionic composition of freshwater systems are highly variable (e.g. heavy metals) and have a strong impact on toxicity and biological uptake dynamics of charged chemicals, yet detailed data on ionic water composition are rarely reported. It would be good include these data

on the Na, K, Ca, Mg and Cl composition of experimental water, either in the methods text or as a table.

L118 - What was the justification for choosing this species of ostracod? E.g. highly abundant in nature? Ecologically important? Previous research on this species? Model organism?

L139-140 - How was temperature measured (e.g. what instrument)? No temperature data are reported for the study either. I doubt the treatment temperatures were exactly 27.0 and 31.0 °C throughout the 48-hour experiment. Temperature mean +/- standard error should be reported for each treatment.

L158 - The text shifts to first person here. It is better to maintain third person throughout as the authors have maintained earlier in the text. Also, just an observation, the author uses the word 'we' at many points throughout the text which is counterintuitive to me as only one author is listed on the manuscript!

L164-165 - This sentence doesn't make sense to me, please rewrite.

L171 - I see that the terms Roundup and glyphosate are used interchangeably between the abstract, introduction, methods and figures. Please be consistent in terminology and add a sentence in the introduction stating how the chemical will be referred to throughout the text.

L172 - I do not understand how the individual impact of temperature can be statistically determined without a temperature treatment containing no chemical.

L173-174, L176 - Please cite the R packages used.

Results

The data is well presented, but I have made suggestions to improve visual guidance to the reader. As mentioned in my previous comments, I don't believe the statistics are possible given the experimental design, and I would suggest the author either modifies the statistical approach and results or includes a true temperature control (without chemical exposure).

L180-181 - Again, I do not see how the individual actions of temperature and chemical can be ascertained.

Figures 2 & 3 - I see no issues with the figures as presented, but it is difficult to see how the dose response curves differ between the treatments. I would suggest combining these results into one figure with each treatment separated by colour, shape and/or line pattern, even with a mark or paired y axis horizontal line and x axis vertical line directing readers to the LC50 values for each treatment.

Table 1 - I don't understand the p values here. Is this comparing if the LC50 value is significantly different from 0? Or is it comparing if the temperature treatments are statistically significant from each other? If the latter is correct, there should only be 1 p value. In fact, I don't see why p values are included in this table at all, since they are given in table 2.

Table 2 - Again, please refer to my earlier comments about the individual effects of chemical and temperature. Additionally, there are inconsistencies in the number of decimal places for these values. Please be consistent with how many significant figures are given.

Discussion

The first half of the discussion is well written and the comparison of LC50 values with other species is interesting and insightful. I would like to see a bit more discussion on the possible mechanisms of the temperature * toxicity interaction. The author mentions lower environmental oxygen, and briefly energetics. However given that I am sure there is plenty literature on chemical LC50 and temperature in a range of invertebrate species (the author cites some of these studies), I think the

manuscript would benefit from a deeper explanation of the physiological mechanisms causing elevated toxicity. For example, elevated metabolic rate at high temperature perhaps causes faster uptake rates of chemicals from the environment into the organisms and ultimately, higher chemical concentrations in the cells at 48 hours exposure. Linked to this, I would also like to see some information on the toxicity mechanisms of the chemical(s) in Roundup. Even if brief, this would help readers understand or perhaps speculate themselves as to why Roundup toxicity increases at higher temperatures in ostracods.

L205-206 - First person is used again I would suggest maintaining third person for consistency.

L209-210 - It would be useful to also include a relative measure of this increase: e.g. a 12 % increase.

L214 - Add the word 'the' before 'active'.

L215 - This is an important point about Roundup vs glyphosate toxicity. Given this research by Tu et al 2001, the author must be clear in referring to the chemical in this study as Roundup and not Glyphosate (as they have done) throughout the manuscript.

L220-221 - Poor grammar, please rewrite this sentence.

L222 - Add Latin name for Fairy shrimps.

L232 - Change 'on pharmaceuticals' to 'to pharmaceuticals'.

L234-235 - Grammar: This sentence needs rewriting.

L232-239: These references would have been useful to include in the Introduction to build the case for the hypothesis.

Conclusions

The conclusions are only partially valid given my interpretation of the experimental design. Without a true temperature control, one cannot confidently claim there is a true interaction effect. It might be that more ostracods die in the high temperature treatment due solely to high temperature alone. Should the author make the necessary changes to the experimental design or statistical analyses, I believe the overall conclusions of this study will be more strongly supported.

Reviewer 2

Comments for the author

This study investigates the putative interactive effects of temperature on the acute sensitivity of the ostracod *Cypridopsis africana* to a commonly used glyphosate-based herbicide (Roundup). By combining two ecologically relevant temperature scenarios (27 °C and 31 °C) with a gradient of herbicide concentrations, the manuscript addresses a timely and relevant question in the context of climate change and ecological risk assessment. The experimental design is generally sound, replication is high, and the main result (a reduction in survival probability at elevated temperature) seems to be well supported by the data. Overall, the study provides useful information for understanding how warming may modulate chemical toxicity in temporary pond organisms. However, several aspects of the manuscript require clarification and improvement, particularly those regarding temperature acclimation procedures, the reporting and interpretation of toxicity metrics (LC50), and the transparency of data presentation. In addition, clearer reporting of water quality parameters, statistical results, and biological interpretation of the temperature by herbicide interaction would substantially strengthen the manuscript's reproducibility and ecological relevance. I also suggest some changes in the title to improve accuracy.

Major comments:

Acclimation to temperature. Please clarify whether ostracods were acclimated to the two temperature treatments prior to exposure, or whether individuals were transferred abruptly from one temperature to another. A sudden +4 °C increase could itself act as a stressor and should be either described explicitly or discussed as a potential contributing factor. Please also confirm control survival at 31 °C and whether you found differences in survival 24 hours post-exposure (please include this information as Figures or Tables either in the main text or as Supplementary Material).

Feeding and water quality. Please state explicitly whether organisms were fed during the 48-h acute test. In addition, while basic water chemistry is reported, the stated analytical detection limit for glyphosate (0.01 ng/L) appears unrealistically low and is likely a unit error (possibly mg/L). Please revise and clarify the analytical detection limit used. Also: did conductivity or other water parameters vary across treatments or temperatures?

GLMM results clarity. The statistical approach (GLMM with binomial error) is appropriate, but Table 2 is confusingly formatted. In particular, the z-value for the interaction term appears incorrect (reported as 0.0000189). Please revise the table and clearly state in the Results that a significant interaction between temperature and Roundup concentration indicates a steeper mortality response at 31 °C 48 hours post-exposure. A brief biological interpretation of model coefficients would improve clarity.

LC50 calculation and interpretation It should be made explicit that LC50 values were estimated by dose-response modelling and do not correspond to experimentally tested concentrations. Please clarify what the "±" values represent (SE or confidence intervals), and preferably report LC50 values with 95% confidence intervals. The current wording may mislead readers into interpreting LC50 values as actual treatment levels. See also below comments on data presentation.

Data presentation. Figures 2 and 3 only show fitted dose-response curves without displaying observed variability. Please report mean mortality (or survival) at each tested concentration together with an appropriate measure of uncertainty (binomial SE or 95% CI). This could be achieved by adding observed data points and error bars to the existing figures, presenting both temperatures in a single figure, and/or including a table summarising mortality/survival per concentration and temperature. In particular, it would be important to report explicitly whether survival remains 100% up to 100 mg/L a.i., as suggested by the curves.

Comment on the title: I suggest revising the title to better reflect the nature of the response variable and the scope of the study. A possibility would be "increasing temperature increases acute toxicity of a glyphosate-based herbicide (Roundup) in the ostracod *Cypridopsis africana*." This formulation more accurately reflects that the study quantifies acute toxicity rather than "tolerance," which may imply physiological or evolutionary adaptation. In addition, I recommend removing the reference to "temporary pond ecosystems" from the title, as the experiment was conducted under controlled laboratory conditions and does not explicitly test ecosystem-level processes or habitat-specific mechanisms. The ecological context can be more appropriately developed in the Introduction and Discussion.

Specific comments by line

Lines 39-40: This sentence restates the main result; please rephrase to avoid presenting it as an additional finding.

Line 67: Do you mean "the most commonly used glyphosate-based herbicide in Tanzania"? Please clarify.

Line 86: Please revise this sentence, as it currently reads awkwardly.

Lines 101-106: Please add a brief statement summarising the main research question and hypothesis. As this is a single-author paper, use "I" instead of "we" throughout.

Lines 110-111: Please provide more detail on the chemical composition of the Roundup formulation used, as formulations can vary across countries.

Lines 117-130: If known, please indicate whether the study population had prior exposure to Roundup. Consider re-ordering this section to first describe the study species, followed by exposure preparation and then realised concentration determination.

Lines 204-210: This section would fit better at the end of the Introduction. The Discussion could begin with a brief summary of the main findings.

Lines 212-225: The comparison with *Daphnia magna* is appropriate, but the Discussion would benefit from a deeper treatment of potential mechanisms underlying the temperature by herbicide interaction (e.g. metabolism, oxidative stress, physiological trade-offs) and from placing the results more clearly within current climate-change ecotoxicology literature (including those investigating the effects of non-lethal levels of some stressors -herbicides but also others- on animal life histories and physiology).

Lines 229-230: Please clarify why reduced dissolved oxygen is proposed as a mechanism (I guess elevated metabolism linked to temperature), or alternatively mention other plausible mechanisms.

Very minor comments and typos

Line 55: "is in partly attributed to" should be "is partly attributed to".

Lines 71-72: insert "with" in "interfering with the production...", and correct "et sl." to "et al."

Lines 73-74: consider rephrasing the sentence on oxidative stress for clarity.

Lines 234-236: sentence is grammatically incorrect, please revise.

Reviewer's Responses to Questions

Experimental quality

Does each figure have the proper controls?

If 'No', please indicate reasons in Comments for Author box below.

Reviewer #1:

- No

Reviewer #2:

- Yes

Were the data analyzed using appropriate statistical tests?

If 'No', please indicate reasons in Comments for Author box below.

Reviewer #1:

- No

Reviewer #2:

- Yes

Reproducibility

Were experiments performed using adequate number of biological replicates?

If 'No', please indicate reasons in Comments for Author box below.

Reviewer #1:

- Yes

Reviewer #2:

- Yes

Does the methods section provide sufficient detail to permit reproducibility?

If 'No', please indicate reasons in Comments for Author box below.

Reviewer #1:

- No

Reviewer #2:

- No

Completeness

Are the manuscript's conclusions supported by the data?

If 'No', please indicate reasons in Comments for Author box below.

Reviewer #1:

- Yes

Reviewer #2:

- Yes

Scholarship

Do the authors cite and discuss the merits of data that would argue for and against their conclusion?

If 'No', please indicate reasons in Comments for Author box below.

Reviewer #1:

- Yes

Reviewer #2:

- Yes

Does the manuscript title & abstract accurately reflect the contents of the manuscript, without hyperbole?

If 'No', please indicate reasons in Comments for Author box below.

Reviewer #1:

- Yes

Reviewer #2:

- No

First revisionAuthor response to reviewers' comments

Dear Editor,

Please find enclosed a revised manuscript titled: **“Increasing temperature exacerbates acute toxicity effects of a glyphosate-based herbicide (Roundup) on the ostracod *Cypridopsis africana*”**. I thank you for an opportunity to revise and submit this manuscript for further consideration. I also appreciate the thorough review effort of two reviewers whose comments have further improved the work. On the table below, I provide responses to all comments given. I hope that the clarifications provided and modifications made in the corresponding manuscript sufficiently address the issues raised and make this manuscript acceptable for publication.

Comment	Response
Reviewer 1	
The hypothesis is not fully justified: At the end of the introduction the author states that they expect increasing temperature to increase Roundup toxicity, but the rationale for this expectation is missing. Why is the hypothesis that it will increase toxicity and not decrease? What has other research shown, and why does the author hypothesize this direction?	Thank you for this observation. I have added a justification and re-stated the hypothesis. Please see lines 99 - 105.

Terminology of Roundup vs glyphosate - these terms are used interchangeably throughout the title, abstract, introduction, methods and figures and should be kept consistent. Given the statement in the discussion about Roundup vs glyphosate toxicity (L215), I would strongly suggest sticking to the term 'Roundup' throughout.	Agreed, I have revised the entire manuscript to ensure consistency in terminology use. Please see lines 35, 39, 215.
Methods missing description of temperature control - There is no explanation for how temperature was controlled, manipulated or measured in the study, and crucially, no temperature data from the experiment are reported. Additionally, no explanation is given as to why + 4 °C was used as the high temperature treatment. Might this already be a lethal temperature for this species? Or is it within the temperature range found in temporary ponds?	I thank you for this comment. Indeed, in standardized acute toxicity test, guidelines require having controls to aid inference of chemical impact of test species. I included controls both at 27 °C and 31 °C. It is, moreover, required by OECD guidelines that, survival in controls be >80%. In this study, no mortality was observed in controls at tested temperatures. Please see lines 138 - 140, 147 - 148 and Figure 1.
Experimental design and statistics - This in my opinion is the biggest flaw in the study. The author does not employ a fully crossed experimental design (independent warming treatment, independent chemical treatment and the two combined equalling 4 treatments). As	I thank you once again for this observation. The experimental design used allows stated hypothesis testing and subsequent data interpretation. In the setup used, an independent warming could be analyzed using controls
such there is no true control for temperature and it is not known whether the increased mortality at high temperature is due solely to temperature, or an interaction between temperature and chemical, potentially calling the findings and conclusions into question. As a result, the chosen statistics are invalid, and the individual effects of temperature and chemical cannot be statistically determined. This makes statistical results in table 2 void. The author should either repeat the study with a true temperature control or change the statistical analyses to a 1-level mixed effects model with warming/chemical exposure as the treatment.	(where no Roundup was added). An independent chemical treatment was included as a gradient of Roundup concentrations at 27 °C and 31 °C while the interactive effects of the two stressors were determined using an interaction term in generalized linear models. Moreover, the endpoint of interest was mortality to be reported as LC50. This can be calculated through dose- response curves and not derived through comparisons across test concentrations. Please see Figure 1, Figure 3 and Table 2.

Mechanistic explanation missing in the discussion - The discussion would greatly benefit from a deeper explanation for possible biological or physiological processes behind the findings as well as the toxicity mechanism of the chemicals present in Roundup. This information is key for the discussion to stimulate further thought and debate among readers, particularly those such as myself who are not familiar with the toxicity mechanism of this specific chemical.	Agreed, a separate paragraph has been added in the discussion section. Here, further explanations on interactive effects Roundup and temperature are provided along with mechanistic explanations. Please see lines 252-264.
L33 and 39 - It is not clear to me what is meant by 'temporary' pond ecosystem in this sentence. This should be briefly defined in the abstract, or this detail omitted in the abstract.	Agreed, I defined temporary ponds and provided context of the ecological importance. See lines 86 - 98.
L35 - Roundup should be briefly defined and explained in the abstract.	Agreed, I added a statement on Roundup, please see lines 35 - 36.
Well written and clearly lays out the research question and its importance and timeliness. What is missing however, is why the author hypothesises temperature to increase sensitivity to chemical exposure. Has this been done before? Or is this the first time that chemical exposure LC50 has been determined in an invertebrate at different temperatures? (I highly doubt that).	Agreed, I have provided a clear justification and re-stated the hypothesis. Please see lines 99-105. It is true that acute toxicity effect of widely used chemical compounds has been determined in traditional model species, the sensitivity of temporary pond species may, however, be different. The difference, as hypothesized, can be attributed to unpredictability in their habitat characteristics (water temperature, dissolved oxygen, pH) which necessitates rapid growth and little energy investment in counteracting effects of chemicals rendering them more susceptible to chemical exposures.
L55 - Grammar: remove the word 'in'.	Agreed, word removed. Please see line 54
L72 - add the word 'with' after the word 'interfering'.	Agreed, word added, please see line 71
L78 - Stating that rising temperatures alter the water quality is not always strictly true. I would suggest re-phrasing this sentence.	Agreed, sentence re-written to include water quality parameters affected by increasing temperature. Please see lines 77 - 79.
L100 - Grammar: change 'elude' to 'eludes'.	Agreed, error corrected. Please see line 101.
L105-106 - The manuscript would benefit from more robust justification for the hypothesis. Why would LC50 be higher at high temperature? Has this been investigated in other species, other chemicals or other systems?	Agreed, further context has now been provided, please see lines 99 - 105.

In general, the methods are clear and comprehensive, and the level of replication is strong, but some important details and justifications are missing as well as water composition data I believe to be relevant. Unfortunately, I am not convinced by the experimental design and accompanying data analysis. The statistical analyses (and results) presented suggest this experiment is a fully crossed design (eg temperature + chemical + chemical * temperature), which it is not. My understanding is that the author has only 2 treatments: a 48-hour LC50 test at 27 degrees (Fig. 2), and a 48-hour LC50 test at 31 degrees (Fig. 3). This means it is not possible to statistically deduce the isolated impacts of temperature on LC50 independent from chemical exposure. I would expect to see a treatment where binary mortality was recorded at both 27 and 31 degrees with no chemical exposure which would constitute a true control and allow the individual effects of temperature and chemical exposure to be statistically analysed as well as their interaction. I concede that conducting an LC50 test without a chemical present is tricky, but this is important to know if the 4 °C increase in temperature is causing the higher mortality by itself, or whether there is indeed a true interaction between temperature and chemical.	Agreed, I have provided further information on key water quality parameters in the reconstituted media (please see lines 149 -152). I acknowledge that measuring mineral and metal composition of water could have provided further substance to the manuscript, however, I strongly believe they had no effect on test conditions as survival in control treatments remained 100% throughout the exposure duration. Studying the interactive effects of temperature and Roundup was still possible by including an interaction term in GLM models. Here, we used mortality data as response variable and Roundup concentration, temperature and their interaction as fixed variables (Please see lines 179 - 181). Moreover, true controls were added at both temperature (please see Figure 1 and Figure 3).
There is also no description of how temperature was manipulated and controlled in this study. Were experiments conducted in stable, temperature-controlled rooms? Or in water baths heated using immersion heaters?. Also	Thank you for this observation. Further details on how temperature was controlled have now been provided (please see line 151 - 152).
why was + 4 °C degrees of warming chosen? Is this based on IPCC end of century predictions or based on previous work?	
L110 - Was the chemical purchased as a solid or solution?	Roundup was purchased in liquid form. Please see line 126

L114 - Why were these concentrations chosen and how do they match with observed concentrations in the field or previous research?	Thank you for this comment. Before setting a full-scale acute toxicity tests, several small range finding experiments were conducted to define the range to which test organisms were responding to (see lines 136 - 137). While it is true that concentrations tested are significantly higher than the currently reported environmental levels, similar levels may be reached in temporary ponds in peak farm preparation seasons, especially in developing countries where chemical use is unregulated.
L116 - I would suggest also including the salinity of this water (e.g. in ppt) for reference. Furthermore, the ionic composition of freshwater systems are highly variable (e.g. heavy metals) and have a strong impact on toxicity and biological uptake dynamics of charged chemicals, yet detailed data on ionic water composition are rarely reported. It would be good include these data on the Na, K, Ca, Mg and Cl composition of experimental water, either in the methods text or as a table.	Thank you for this suggestion. I used Instant Ocean Salt to reconstitute the exposure media to 490 $\mu\text{S}/\text{cm}$ matching with the electric conductivity levels of temporary ponds. Among other trace minerals, Instant Ocean Salt has Na, K, Ca, Mg and Cl. While data on Na, K, Ca, Mg and Cl composition of experimental water would indeed be useful, I am convinced that they were added at desirable levels following an observed 100% survival of test organisms in control treatment throughout the experimental duration.
L118 - What was the justification for choosing this species of ostracod? E.g. highly abundant in nature? Ecologically important? Previous research on this species? Model organism?	Thank you for the comment. Ostracods were selected as test species due to their key ecological role as filter feeders, their omnipresence in temporary ponds and they are easy to breed and maintain in the laboratory. Please see lines 109 - 111.
L139-140 - How was temperature measured (e.g. what instrument)? No temperature data are reported for the study either. I doubt the treatment temperatures were exactly 27.0 and 31.0 °C throughout the 48-hour experiment. Temperature mean +/- standard error should be reported for each treatment.	Thank you for this comment. Indeed, temperature varied by ± 1 °C in all conditions (please see lines 151 -152). Used thermostat heaters had a temperature display for monitoring water temperature throughout the exposure duration.
L158 - The text shifts to first person here. It is better to maintain third person throughout as the authors have maintained earlier in the text. Also, just an observation, the author uses the word 'we' at many points throughout the text which is	Agreed, this has been corrected throughout the manuscript. Please see lines 102, 237, 268 and 272.
counterintuitive to me as only one author is listed on the manuscript!	
L164-165 - This sentence doesn't make sense to me, please rewrite.	Agreed, sentence rewritten, please see lines 173-174.

L171 - I see that the terms Roundup and glyphosate are used interchangeably between the abstract, introduction, methods and figures. Please be consistent in terminology and add a sentence in the introduction stating how the chemical will be referred to throughout the text.	Agreed, this has been changed throughout the manuscript. Roundup has been used for consistency. Please see, among others, lines 35, 39, 215.
L172 - I do not understand how the individual impact of temperature can be statistically determined without a temperature treatment containing no chemical.	Thank you for this comment. A treatment with no chemical was added both at 27 °C and 31 °C. As expected, no mortality was observed. Please see figures 1 and 2.
L173-174, L176 - Please cite the R packages used	Agreed, packages used have been cited. Please see lines 182 - 183.
The data is well presented, but I have made suggestions to improve visual guidance to the reader. As mentioned in my previous comments, I don't believe the statistics are possible given the experimental design, and I would suggest the author either modifies the statistical approach and results or includes a true temperature control (without chemical exposure).	Thank you once again for this observation. I have now included a figure to show mortalities at different concentrations (please see figure 3). Controls (treatments with no chemicals) were part of the design (please see lines 140, 147, Figures 1 and 3).
L180-181 - Again, I do not see how the individual actions of temperature and chemical can be ascertained.	Thank you for this observation. Individual effects of temperature, Roundup and their interaction could be determined using GLM with acute data from both 27 °C and 31 °C. Please see lines 179 - 183.
Figures 2 & 3 - I see no issues with the figures as presented, but it is difficult to see how the dose response curves differ between the treatments. I would suggest combining these results into one figure with each treatment separated by colour, shape and/or line pattern, even with a mark or paired y axis horizontal line and x axis vertical line directing readers to the LC50 values for each treatment.	Agreed, I combined the figures and added a dotted line to indicate the LC50. Please see figure 2.
Table 1 - I don't understand the p values here. Is this comparing if the LC50 value is significantly different from 0? Or is it comparing if the temperature treatments are statistically significant from each other? If the latter is correct, there should only be 1 p value. In fact, I don't see why p values are included in this table at all, since they are given in table 2.	Thank you for the comment. Indeed, the p values compares if the intercept of the dose-response curve is significantly different from zero. Agreed, I have removed the p values.
Table 2 - Again, please refer to my earlier comments about the individual effects of chemical and temperature. Additionally, there are inconsistencies in the number of decimal places for these values. Please be consistent with how many significant figures are given.	Thank you for this comment. Individual effects of temperature can be determined using GLM models with temperature being a fixed factor. Agreed, I have edited the table to ensure consistency in significant figures.

The first half of the discussion is well written and the comparison of LC50 values with other species is interesting and insightful. I would like to see a bit more discussion on the possible mechanisms of the temperature * toxicity interaction. The author mentions lower environmental oxygen, and briefly energetics. However given that I am sure there is plenty literature on chemical LC50 and temperature in a range of invertebrate species (the author cites some of these studies), I think the manuscript would benefit from a deeper explanation of the physiological mechanisms causing elevated toxicity. For example, elevated metabolic rate at high temperature perhaps causes faster uptake rates of chemicals from the environment into the organisms and ultimately, higher chemical concentrations in the cells at 48 hours exposure. Linked to this, I would also like to see some information on the toxicity mechanisms of the chemical(s) in Roundup. Even if brief, this would help readers understand or perhaps speculate themselves as to why Roundup toxicity increases at higher temperatures in ostracods.	Agreed, I have now added a paragraph in the discussion section expanding further on the interactive effects of temperature and chemicals with focus on temporary ponds. Please see lines 252 - 264.
L205-206 - First person is used again I would suggest maintaining third person for consistency.	Agreed, I have revised the entire manuscript to correct this.
L209-210 - It would be useful to also include a relative measure of this increase: e.g. a 12 % increase.	Agreed, this has now been included, please see line 219.
L214 - Add the word 'the' before 'active'.	Agreed, "the" has been added, please see line 223.
L215 - This is an important point about Roundup vs glyphosate toxicity. Given this research by Tu et al 2001, the author must be clear in referring to the chemical in this study as Roundup and not Glyphosate (as they have done) throughout the manuscript.	Agreed, I revised the whole manuscript to refer to Roundup and not glyphosate.
L220-221 - Poor grammar, please rewrite this sentence.	Agreed, the sentence has been rewritten, please see lines 229 - 230.
L222 - Add Latin name for Fairy shrimps.	Agreed, scientific name has been added, please see line 232.
L232 - Change 'on pharmaceuticals' to 'to pharmaceuticals'.	Agreed, I made the suggested change. Please see line 244.
L234-235 - Grammar: This sentence needs rewriting.	Agreed, the sentence was rewritten. Please see lines 245 - 246.

L232-239: These references would have been useful to include in the Introduction to build the case for the hypothesis.	Thank you for the comment. Hypothesis has now been justified and clearly stated. Please see lines 99 - 105.
The conclusions are only partially valid given my interpretation of the experimental design. Without a true temperature control, one cannot confidently claim there is a true interaction effect. It might be that more ostracods die in the high temperature treatment due solely to high temperature alone. Should the author make the necessary changes to the experimental design or statistical analyses, I believe the overall conclusions of this study will be more strongly supported.	Thank you for this comment. We observed no mortality in control treatments both at 27 °C and 31 °C (see lines 152 - 153, figure 2 and 3). Mortalities were therefore due to Roundup and interaction between Roundup and temperature.
Reviewer 2	
Acclimation to temperature. Please clarify whether ostracods were acclimated to the two temperature treatments prior to exposure, or whether individuals were transferred abruptly from one temperature to another. A sudden +4 °C increase could itself act as a stressor and should be either described explicitly or discussed as a potential contributing factor. Please also confirm control survival at 31 °C and whether you found differences in survival 24 hours post-exposure (please include this information as Figures or Tables either in the main text or as Supplementary Material).	Thank you for this observation. Indeed, ostracods were acclimated prior to their transfer to 31 °C. In all controls, at 27 °C and 31 °C, survival remained 100% and mortality resulted by the addition of Roundup at different concentrations. I added a violin plot showing survival in controls at the two temperature treatments (please see figure 3).
Feeding and water quality. Please state explicitly whether organisms were fed during the 48-h acute test. In addition, while basic water chemistry is reported, the stated analytical detection limit for glyphosate (0.01 ng/L) appears unrealistically low and is likely a unit error (possibly mg/L). Please revise and clarify the analytical detection limit used. Also: did conductivity or other water parameters vary across treatments or temperatures?	Thank you for the comment. As stipulated in the OECD guideline, test species were not fed in the 48 hours of exposure. Agreed, I corrected a mistake related to analytical detection limit. Core water quality parameters did not vary across temperature treatments (please see line 176).
GLMM results clarity. The statistical approach (GLMM with binomial error) is appropriate, but Table 2 is confusingly formatted. In particular, the z-value for the interaction term appears incorrect (reported as 0.0000189). Please revise the table and clearly state in the Results that a significant interaction between temperature and Roundup concentration indicates a steeper mortality response at 31 °C 48 hours post- exposure. A brief biological interpretation of model coefficients would improve clarity.	Agreed, the table has been revised. Additionally, further clarification has been provided as suggested (see lines 194 - 195).

LC50 calculation and interpretation. It should be made explicit that LC50 values were estimated by dose-response modelling and do not correspond to experimentally tested concentrations. Please clarify what the "±" values represent (SE or confidence intervals), and preferably report LC50 values with 95% confidence intervals. The current wording may mislead readers into interpreting LC50 values as actual treatment levels. See also below comments on data presentation.	Agreed, further context has been provided (see lines 187 - 188. Standard error provides the reliability of the LC50. A smaller SE suggests a more reliable estimate, while a larger SE indicates more uncertainty in the reported LC50. Here, SE values are -7% of the LC50 which is within an acceptable level of 10%.
Data presentation. Figures 2 and 3 only show fitted dose-response curves without displaying observed variability. Please report mean mortality (or survival) at each tested concentration together with an appropriate measure of uncertainty (binomial SE or 95% CI). This could be achieved by adding observed data points and error bars to the existing figures, presenting both temperatures in a single figure, and/or including a table summarizing mortality/survival per concentration and temperature. In particular, it would be important to report explicitly whether survival remains 100% up to 100 mg/L a.i., as suggested by the curves.	Agreed, I added a violin plot which better presents the variability. Please see figure 3
Comment on the title: I suggest revising the title to better reflect the nature of the response variable and the scope of the study. A possibility would be "increasing temperature increases acute toxicity of a glyphosate-based herbicide (Roundup) in the ostracod Cypridopsis africana." This formulation more accurately reflects that the study quantifies acute toxicity rather than "tolerance," which may imply	Agreed, the title was changed as suggested, please see lines 1 - 2.
physiological or evolutionary adaptation. In addition, I recommend removing the reference to "temporary pond ecosystems" from the title, as the experiment was conducted under controlled laboratory conditions and does not explicitly test ecosystem-level processes or habitat-specific mechanisms. The ecological context can be more appropriately developed in the Introduction and Discussion.	
Lines 39-40: This sentence restates the main result; please rephrase to avoid presenting it as an additional finding.	Agreed, the statement has been rephrased. Please see lines 39 - 40.

Line 67: Do you mean "the most commonly used glyphosate-based herbicide in Tanzania"? Please clarify.	Indeed, the sentence was corrected, see line 66
Line 86: Please revise this sentence, as it currently reads awkwardly.	Agreed, the sentence has been rewritten. Please see lines 86.
Lines 101-106: Please add a brief statement summarising the main research question and hypothesis. As this is a single-author paper, use "I" instead of "we" throughout.	Agreed, I have now added the main study objective and hypothesis along with a justification. Please see lines 99 - 105.
Lines 110-111: Please provide more detail on the chemical composition of the Roundup formulation used, as formulations can vary across countries.	Agreed, the chemical composition of Roundup has now been added. Please see lines 127 - 129.
Lines 117-130: If known, please indicate whether the study population had prior exposure to Roundup. Consider re-ordering this section to first describe the study species, followed by exposure preparation and then realised concentration determination.	Thank you for this comment. In sediments collection (for ostracod resting eggs), I purposely sampled ponds further away from agricultural areas. Majority of the ponds were in pastoral areas where nomadic pastoralists occasionally graze their cattle. I have re-ordered the section as suggested.
Lines 204-210: This section would fit better at the end of the Introduction. The Discussion could begin with a brief summary of the main findings.	Agree, the section has been revised and merged as suggested, see line 99 - 105.
Lines 212-225: The comparison with Daphnia magna is appropriate, but the Discussion would benefit from a deeper treatment of potential mechanisms underlying the temperature by herbicide interaction (e.g. metabolism, oxidative stress, physiological trade-offs) and from placing the results more clearly within current climate-change ecotoxicology literature (including those investigating the effects of non-lethal levels of some stressors -herbicides but also others- on animal life histories and physiology).	Agreed, I have now added a paragraph in the discussion section expanding further on the interactive effects of temperature and chemicals with focus on temporary ponds. Please see lines 252 - 264.
Lines 229-230: Please clarify why reduced dissolved oxygen is proposed as a mechanism (I guess elevated metabolism linked to temperature), or alternatively mention other plausible mechanisms.	Agreed, the statement has been replaced with more plausible mechanism (see lines 239 - 240).
Line 55: "is in partly attributed to" should be "is partly attributed to".	Agreed, correction made, please see line 54.
Lines 71-72: insert "with" in "interfering with the production...", and correct "et sl." To "et al."	Agreed, suggested changes were made, see lines 70 - 72.
Lines 73-74: consider rephrasing the sentence on oxidative stress for clarity.	Agreed, the sentence was rephrased. Please lines 72 - 74.
Lines 234-236: sentence is grammatically incorrect, please revise.	Agreed, the sentence was rewritten, please see lines 245 - 246.

Second decision letter

MS ID#: bio.062487R1

MS Title: Increasing temperature exacerbates acute toxicity effects of a glyphosate-based herbicide (Roundup) on the ostracod *Cypridopsis africana*

Authors: Yusuph Aron Kafula

I have now reached a decision on the above manuscript. As you will see, the reviewers appreciated the changes you made to the manuscript (see reports below). However, they had a few additional concerns that need to be addressed. I would like to see all of the points addressed, but in particular the concern about how the data are presented in Figure 3 is the most important. I hope that you will be able to make these changes, because we would like to be able to accept your paper.

At this stage, we also ask you to ensure your manuscript complies with our formatting guidelines - please see our manuscript preparation guidelines for details. Provided you are able to fully address the referees' comments, we are positive about publication of your paper (we accept over 95% of revision submissions) and therefore hope you won't mind any extra work involved in reformatting your manuscript at this point.

Please upload both a 'clean' version of your Word file, along with a highlighted version clearly showing where you have made changes in the revised manuscript. Please avoid using 'Track changes' in Word files as these are lost in PDF conversion.

I should be grateful if you would also provide a point-by-point response detailing how you have dealt with the points raised by the reviewers in the 'Response to Reviewers' box. Please attend to all of the reviewers' comments. If you do not agree with any of their criticisms or suggestions please explain clearly why this is so.

Reviewer 1

No further comments. Please see response to reviewer questionnaire.

Reviewer 2

Comments for the authors

I would like to thank the authors for the careful and detailed responses to the previous comments. The manuscript has clearly improved in structure, clarity, and interpretation, and several key points raised in the first review round have been addressed satisfactorily. That said, there are still a few aspects that would benefit from further clarification.

First, regarding temperature acclimation, it is now stated that organisms were acclimated prior to transfer to 31 °C and that control survival was 100% at both temperatures. However, these details need to be reported much more precisely in the Methods. The manuscript should clearly specify how acclimation was conducted (gradual versus abrupt transfer), over what time period, and at which life stage. A +4 °C increase can itself represent a physiological stressor, potentially affecting metabolism and short-term performance, and explicit methodological detail is essential for reproducibility and correct interpretation of the temperature × toxicity interaction.

Second, concerning feeding during the 48-hour exposure, I appreciate the clarification that organisms were not fed in accordance with OECD guidelines. Nevertheless, beyond methodological conventions, this aspect deserves brief discussion. Elevated temperature increases metabolic

demand, and short-term fasting may influence energetic status and sensitivity to toxicants. Acknowledging this potential biological effect would strengthen the interpretation of the results.

Finally, while the addition of violin plots improves visualization, the presentation of observed data remains insufficiently explicit. Violin plots do not replace clear reporting of observed mortality (or survival) per concentration and temperature together with appropriate measures of uncertainty (e.g. binomial SE or 95% confidence intervals). Including observed data points with error bars or a summary table of mortality/survival across treatments would greatly improve transparency and allow readers to better assess variability and dose-response patterns.

Reviewer's Responses to Questions

Experimental quality

Does each figure have the proper controls?

If 'No', please indicate reasons in Comments for Author box below.

Reviewer #1:

- Yes

Reviewer #2:

- Yes

Were the data analyzed using appropriate statistical tests?

If 'No', please indicate reasons in Comments for Author box below.

Reviewer #1:

- Yes

Reviewer #2:

- Yes

Reproducibility

Were experiments performed using adequate number of biological replicates?

If 'No', please indicate reasons in Comments for Author box below.

Reviewer #1:

- Yes

Reviewer #2:

- Yes

Does the methods section provide sufficient detail to permit reproducibility?

If 'No', please indicate reasons in Comments for Author box below.

Reviewer #1:

- No

Reviewer #2:

- No

Completeness

Are the manuscript's conclusions supported by the data?

If 'No', please indicate reasons in Comments for Author box below.

Reviewer #1:

- Yes

Reviewer #2:

- Yes

Scholarship

Do the authors cite and discuss the merits of data that would argue for and against their conclusion?

If 'No', please indicate reasons in Comments for Author box below.

Reviewer #1:

- Yes

Reviewer #2:

- Yes

Does the manuscript title & abstract accurately reflect the contents of the manuscript, without hyperbole?

If 'No', please indicate reasons in Comments for Author box below.

Reviewer #1:

- Yes

Reviewer #2:

- Yes
-

Second revisionAuthor response to reviewers' comments

Dear Editor,

Please find enclosed a revised manuscript titled: “**Increasing temperature exacerbates acute toxicity effects of a glyphosate-based herbicide (Roundup) on the ostracod *Cypridopsis africana***”. I thank you for yet another opportunity to revise and submit this manuscript for consideration in Biology Open. I once again appreciate the thorough review effort of two reviewers whose comments have substantially improved the work. On the table below, I provide responses to comments given by reviewer 2. I hope that the clarifications provided and modifications made in the corresponding manuscript sufficiently address additional issues raised and make this manuscript acceptable for publication.

Comment	Response
Reviewer 2	
Regarding temperature acclimation, it is now stated that organisms were acclimated prior to transfer to 31 °C and that control survival was 100% at both temperatures. However, these details need to be reported much more precisely in the Methods. The manuscript should clearly specify how acclimation was conducted (gradual versus abrupt transfer), over what time period, and at which life stage. A +4 °C increase can itself represent a physiological stressor, potentially affecting metabolism and short-term performance, and explicit methodological detail is essential for reproducibility and correct interpretation of the temperature × toxicity interaction.	Thank you for this comment. Agreed, I have now added details of how acclimation was done on test groups assigned to 31 °C. Please see lines 124 - 127.

Concerning feeding during the 48-hour exposure, I appreciate the clarification that organisms were not fed in accordance with OECD guidelines. Nevertheless, beyond methodological conventions, this aspect deserves brief discussion. Elevated temperature increases metabolic demand, and short-term fasting may influence energetic status and sensitivity to toxicants. Acknowledging this potential biological effect would strengthen the interpretation of the results.	Thank you for this comment, I added a rationale of why feeding in not recommended. Please see lines 149 - 150. While I agree discussion on feeding regimes could provide another angle of explaining my findings, this could have diluted the key message on the role of temperature on Roundup toxicity. Moreover, accounting on the effect of feeding would require a separate treatment to be able to clearly pinpoint its interplay with temperature and Roundup.
While the addition of violin plots improves visualization, the presentation of observed data remains insufficiently explicit. Violin plots do not replace clear reporting of observed mortality (or survival) per concentration and temperature together with appropriate measures of uncertainty (e.g. binomial SE or 95% confidence intervals). Including observed data points with error bars or a summary table of mortality/survival across treatments would greatly improve transparency and allow readers to better assess variability and dose-response patterns.	Agreed, I have replaced the violin plots with error bars. Please see figure 2.

Third decision letter

MS ID#: bio.062487R2

MS Title: Increasing temperature exacerbates acute toxicity effects of a glyphosate-based herbicide (Roundup) on the ostracod *Cypridopsis africana*

Authors: Yusuph Aron Kafula

Thank you for the additional changes to the manuscript. I apologize for this, but I have one more request regarding Figure 3. The requirement for Biology Open is to show individual data points along with the error bars. "For best practice and transparency, and to allow better assessment of the quality of the data and whether the data support the conclusions, BiO requires that you use graphs that allow the reader to see the true data spread (unless an $n = 1$ is stated). For bar graphs with error bars, individual data points are required."

<https://journals.biologists.com/bio/pages/manuscript-prep#section-4-1>

Could you overlay the individual data points onto the bar graphs? If you are able to complete that, I would be happy to accept the paper. At this stage, we also ask you to ensure your manuscript complies with our formatting guidelines - please see our manuscript preparation guidelines for details. Provided you are able to fully address the referees' comments, we are positive about publication of your paper (we accept over 95% of revision submissions) and therefore hope you won't mind any extra work involved in reformatting your manuscript at this point.

Third revision

Author response to reviewers' comments

Dear Editor,

© 2026. Published by The Company of Biologists under the terms of the Creative Commons Attribution License (<https://creativecommons.org/licenses/by/4.0/>).

Please find enclosed a revised manuscript titled: “**Increasing temperature exacerbates acute toxicity effects of a glyphosate-based herbicide (Roundup) on the ostracod *Cypridopsis africana***”. I thank you for yet another opportunity to revise and submit this manuscript for consideration in Biology Open. On the table below, I provide a response to your comment. I hope that the modifications made on figure 3 and in the corresponding manuscript sufficiently address the additional issue raised and make this manuscript acceptable for publication.

Comment	Response
Editor	
The requirement for Biology Open is to show individual data points along with the error bars. "For best practice and transparency, and to allow better assessment of the quality of the data and whether the data support the conclusions, BiO requires that you use graphs that allow the reader to see the true data spread (unless an n = 1 is stated). For bar graphs with error bars, individual data points are required.	Thank you for this comment. I agree; I have now overlaid individual data points on the error bars at 27 °C and 31 °C. Please see Figure 3.

Fourth decision letter

MS ID#: bio.062487R3

MS Title: Increasing temperature exacerbates acute toxicity effects of a glyphosate-based herbicide (Roundup) on the ostracod *Cypridopsis africana*

Authors: Yusuph Aron Kafula

I am happy to tell you that your manuscript has been accepted for publication in Biology Open, pending our standard publication integrity checks. It was accepted on 3rd March 2026.